# *GsRSS3L*, a Candidate Gene Underlying Soybean Resistance to Seedcoat Mottling Derived from Wild Soybean (*Glycine soja* Sieb. and Zucc)

**DOI:** 10.3390/ijms23147577

**Published:** 2022-07-08

**Authors:** Shuang Song, Jing Wang, Xingqi Yang, Xuan Zhang, Xiuli Xin, Chunyan Liu, Jianan Zou, Xiaofei Cheng, Ning Zhang, Yuxi Hu, Jinhui Wang, Qingshan Chen, Dawei Xin

**Affiliations:** Key Laboratory of Soybean Biology of Chinese Ministry of Education, Key Laboratory of Soybean Biology and Breeding/Genetics of Chinese Agriculture Ministry, College of Science, Northeast Agricultural University, Harbin 150030, China; songshuang.000@163.com (S.S.); wangjing20211029@163.com (J.W.); y13603682439@163.com (X.Y.); zx13194520400@163.com (X.Z.); xinxiuli@163.com (X.X.); cyliucn@126.com (C.L.); zoujianan@neau.edu.cn (J.Z.); xfcheng@neau.edu.cn (X.C.); zn788635@163.com (N.Z.); a01190194@neau.edu.cn (Y.H.); jinhuiwang113@126.com (J.W.)

**Keywords:** SMV, seed coat mottling, chromosome segments substituted lines, resistance

## Abstract

Soybeans are a major crop that produce the best vegetable oil and protein for use in food and beverage products worldwide. However, one of the most well-known viral infections affecting soybeans is the Soybean Mosaic Virus (SMV), a member of the *Potyviridae* family. A crucial method for preventing SMV damage is the breeding of resistant soybean cultivars. Adult resistance and resistance of seedcoat mottling are two types of resistance to SMV. Most studies have focused on adult-plant resistance but not on the resistance to seedcoat mottling. In this study, chromosome segment-substituted lines derived from a cross between Suinong14 (cultivated soybean) and ZYD00006 (wild soybean) were used to identify the chromosome region and candidate genes underlying soybean resistance to seed coat mottling. Herein, two quantitative trait loci (QTLs) were found on chromosome 17, and eighteen genes were found in the QTL region. RNA-seq was used to evaluate the differentially expressed genes (DEGs) among the eighteen genes located in the QTLs. According to the obtained data, variations were observed in the expression of five genes following SMV infection. Furthermore, *Nicotiana benthamiana* was subjected to an *Agrobacterium*-mediated transient expression assay to investigate the role of the five candidate genes in SMV resistance. It has also been revealed that *Glyma.17g238900* encoding a RICE SALT SENSITIVE 3-like protein (RSS3L) can inhibit the multiplication of SMV in *N.*
*benthamiana*. Moreover, two nonsynonymous single-nucleotide polymorphisms (SNPs) were found in the coding sequence of *Glyma.17g238900* derived from the wild soybean ZYD00006 (*GsRSS3L*), and the two amino acid mutants may be associated with SMV resistance. Hence, it has been suggested that *GsRSS3L* confers seedcoat mottling resistance, shedding light on the mechanism of soybean resistance to SMV.

## 1. Introduction

Soybean (*Glycine max* L. Merr.), the most important legume crop in the world, is an essential and dominant source of protein for humans and animals [1]. Various biotic and abiotic stresses could affect the yield and quality of soybeans. Among the biotic stresses, infection by some plant viruses, such as soybean mosaic virus (SMV), bean pod mottle virus (BPMV), soybean vein necrosis virus (SVNV), tobacco ringspot virus (ToRSV), and alfalfa mosaic virus (AMV), causes great damage to soybean production [2,3]. SMV is the most frequent and dangerous virus infecting soybeans, causing production losses of up to 50% [4].

SMV, a member of the family Potyviridae’s genus *Potyvirus*, is a seed- and aphid-borne virus that can be spread through mechanical inoculation [5]. It has a single-stranded positive-sense genome of approximately 10 kb containing a single large open reading frame (ORF), which is translated and cleaved into ten mature proteins (P1, HC-Pro, P3, 6K1, CI, 6K2, VPg, NIa-Pro, Nib, and CP) and a small ORF, which is produced by a frameshift in the P3 cistron and expressed as a fusion protein (P3N-PIPO) [6]. Based on the pathogenicity in different soybean cultivars, SMV isolates have been grouped into different strains, including G1-G7 strains in the United States (US), SC1-SC22 strains in Southern China, and N1-N3 strains in Northeast China [7,8]. In addition to foliar symptoms, such as mottling, mosaic, distortion, and vein necrosis, SMV infection of soybean can cause seed symptoms of seed coat mottling due to the partial suppression of the endogenous RNA silencing pathway of chalcone synthase (CHS) mRNA [9,10].

The use of genetic resistance is the most effective, economical, and environment-friendly strategy against viral disease [11]. To date, multiple independent resistance loci have been identified in various soybean cultivars, such as four dominant loci (*Rsv1*, *Rsv3*, *Rsv4*, and *Rsv5*) conferring resistance against SMV G1-G7 strains in the US and a series of *Rsc* loci conferring resistance against SCs strains in China [2]. The resistance of leaves to SMV infection is known as adult-plant resistance [12]. In addition to adult-plant resistance, Cooper (1966) was the first to report that the soybean cultivar Merit was resistant to seed coat mottling [13], and Hu et al. (1996) proved that the resistance to seed coat mottling and SMV infection was controlled by distinct genes [14]. The genetic analysis revealed that the chromosome loci responsible for soybean seed coat mottling resistance are distinct from the loci responsible for adult-plant resistance [12]. To date, all reported resistance loci or genes confer adult-plant resistance, and no seed coat mottling resistance gene has been fine mapped or cloned.

Crop wild germplasm contributes an invaluable genetic resource for crop improvement [15]. The wild progenitor species of soybean (*Glycine soja* Sieb. and Zucc.), wild soybean (*Glycine soja* Sieb. and Zucc.), has numerous resistance genes to biotic and abiotic stresses with potential applications in soybean breeding [16]. Hence, we developed a population of chromosomal segment substitution lines (CSSLs), with soybean cultivar SN14 as the recurrent parent and wild soybean ZYD00006 as the donor parent to efficiently retrieve beneficial genes from wild soybean [17]. Herein, the responses of the CSSL population to SMV infection were characterized, and then potential genes linked with seedcoat mottling resistance were assessed by combining QTL and RNA-seq analyses. The obtained data revealed the inhibition of SMV multiplication for a candidate gene encoding a RICE SALT SENSITIVE 3-like protein (RSS3L) by *Agrobacterium*-mediated transient expression in *Nicotiana benthamiana*. Our data identified a novel QTL locus and candidate gene underlying soybean resistance to seed coat mottling.

## 2. Results

### 2.1. Response of CSSL Population to SMV

To detect the variation in seed coat mottling induced by SMV N3 strain in CSSL, we identified the phenotype of CSSLs after inoculation with SMV N3. According to the obtained results, mechanical inoculation of SMV N3 strain on CSSL lines caused different symptoms. As such, 9 of the 194 CSSL lines showed hypersensitive necrotic spots on inoculated leaves within 7 days post-inoculation (dpi), while non-inoculated systemic leaves did not show symptoms until 21 dpi (Figure 1a). No virus was detected by RT-PCR in systemic leaves of the nine CSSL lines at 21 dpi, demonstrating their extreme resistance against SMV N3 infection. On SN14 and each of the other 185 CSSL lines, symptoms of severe mosaic and vein necrosis were evident on inoculated leaves within 7 dpi, and chlorosis and crinkling on systemic leaves within 14 to 21 dpi (Figure 1a). Although all 185 CSSL lines were sensitive to SMV N3 infection, the incidence of seed coat mottling ranged from 0% to 100%. (Figure 1b). No seed coat mottling symptoms were observed on any seed from the nine lines with extreme resistance to SMV N3 infection. These findings imply that the nine seed-coat-mottling-resistant lines have both seed coat mottling and adult-plant resistance, and the substituted wild soybean genome should be the primary factor behind the variation in CSSLresistance to SMV N3.

### 2.2. Identification of Genomic Regions Associated with Resistance to Seed Coat Mottling

Resistance to seed coat mottling differs in CSSLs due to differences in chromosomal substituted segments and wild soybean genomic data. Hence, we aimed to evaluate the chromosomal region underlying the phenotypic variation. The genomic regions associated with resistance to seed coat mottling were identified by QTL analysis based on the incidence rates of seed coat mottling of the 185 CSSL lines which could be systemically infected with SMV N3. According to the obtained data, only two QTLs were found on chromosome 17, and both were located at the end of chromosome 17 (Figure 2). Block9566 and Block9910 were the two substituted regions on chromosome 17, with logarithmic odds ratios (LOD) of 2.50 and 7.90, respectively (Table 1). The Block9566 of about 141 kb represented 4.57% of observed phenotypic variation and contained 16 annotated genes according to the reference genome GlymaWm82.a2.v1 annotations (Figure 2; Appendix A). The Block9910 was about 24 kb and contained two annotated genes, *Glyma.17g238900* and *Glyma.**17g239000*, which accounted for 14.66% of observed phenotypic variation (Figure 2; Appendix A).

### 2.3. RNA-Seq Analysis of SN14 Response to SMV Infection

RNA-seq was performed to find the potential genes located in the QTLs to better understand the variation in gene expression caused by SMV N3. The RNA-seq data of SN14 were compared after the infection treatment between SMV N3 and control treatment of PBS buffer at 3, 6, 12, 24, and 72 h post-inoculation (hpi). A total of 30 RNA-Seq libraries were constructed and sequenced. On average, there were 6.5 billion bases and 21.5 million clean reads per library, with a high Q30 (base ratio 30) range of 93.2 to 95.1% (Appendix A). Further analyses identified 4856, 3488, 1236, 478, and 143 differentially expressed genes (DEGs) under SMV infection at 3, 6, 12, 24, and 72 hpi, respectively, among which 2683, 1433, 756, 301, and 130 DEGs were upregulated and 2173, 2055, 480, 177, and 13 DEGs were downregulated (Figure 3a–c). These findings demonstrated that early SMV infection had a significant impact on the expression profiles of global transcripts in soybean. The Kyoto Encyclopedia of Genes and Genomes (KEGG) enrichment analyses revealed that the upregulated DEGs were primarily enriched in the biosynthesis or metabolism of some antioxidative molecules, including glutathione and phenylpropanoid, as well as some pathways of different amino acid and fatty acid metabolism (Appendix A). Moreover, the downregulated DEGs were mainly enriched in photosynthesis-related pathways, indicating the suppression of the photosynthesis process in SMV-infected soybean (Appendix A).

### 2.4. Prediction and qRT-PCR Validation of Candidate Genes

Five DEGs (*Glyma.17g091000*, *Glyma.17g091200*, *Glyma.17g092000*, *Glyma.17g092500*, and *Glyma.17g238900*) were detected in the two regions identified by QTL analysis and chosen as potential candidate genes for resistance to seed coat mottling after combining the results of QTL and RNA-seq analyses (Figure 3d). The expression patterns of the five DEGs were further validated by qRT-PCR and the obtained results showed consistency with the RNA-seq data (Figure 4). *Glyma.17g092000* was upregulated at 3 and 6 hpi with SMV infection. *Glyma.17g091000* and *Glyma.17g238900* were downregulated both at 3 and 6 hpi, while *Glyma.17g091200* and *Glyma.17g092500* were downregulated only at 3 hpi. These findings indicated that the underlined five potential genes within the QTL are involved in the response to SMV N3 strain infection.

### 2.5. Agrobacterium-Mediated Transient Expression of Candidate Genes

*Agrobacterium*-mediated transient co-expression of candidate genes and full-length cDNA infectious clones of SMV in *Nicotiana benthamiana* were carried out to determine the role of the five candidate genes in SMV N3 strain resistance. The findings showed that the other four potential genes, except *Glyma.17g238900*, had no impact on SMV infection or multiplication in *N. benthamiana*. *Glyma.17g238900* encodes a protein consisting of 380 amino acids (aa), which contains an N-terminal domain of basic helix-loop-helix (bHLH) transcription factors while lacking the DNA binding domain. Herein, BLAST analysis revealed that it is homologous to the protein RSS3 encoded by *Os11g0446000* in *Oryza sativa* [18], hence, it was named an RSS3-like (RSS3L) protein.

The sequence of *Glyma.17g238900* from SN14 and ZYD00006, named *GmRSS3L* and *GsRSS3L*, respectively, was amplified and sequenced. There was a total of nine single-nucleotide polymorphisms (SNPs) in the promoter region and four SNPs in the coding sequence, including two synonymous and two nonsynonymous variants (Figure 5a). These two nonsynonymous SNPs led to coded amino acid mutants, one of which is Ser to Phe and the other is Cys to Tyr (Figure 5a). Due to these two amino acid mutants, an α-helix was lacking based on the prediction of protein structure (https://swissmodel.expasy.org/interactive, accessed on 6 May 2022) (Figure 5b). Furthermore, the plant expression vectors of *GmRSS3L* and *GsRSS3L* were constructed and transient co-expressed with the infectious clone of SMV. The results of qRT-PCR showed that the levels of viral RNA were significantly lower in inoculated *N. benthamiana* leaves with transiently expressed *GsRSS3L* than those with *GmRSS3L* and control treatment at 5 and 7 dpi (Figure 5c). The underlined results suggested that *GsRSS3L* inhibits the multiplication of SMV in *N. benthamiana* and it is the candidate gene underlying the resistance to seed coat mottling.

## 3. Discussion

Soybeans are a major crop that produces the best vegetable oil and protein for use in food and beverage products worldwide. The identified QTLs on chromosome 17 underlying resistance to seed coat mottling enable this study to uncover a novel potential resistant locus in soybean. Largely, most of the identified dominant genes with resistance against SMV were located on the chromosome 2, 6, 13, and14, such as *Rsv1*, *Rsv3*, *Rsv4*, *Rsv5*, *Rsc3*, *Rsc5*, *Rsc7*, *Rsc8*, *Rsc12*, *Rsc18,* and *Rsc20* [11,19,20,21,22,23,24,25,26,27,28]. This result demonstrated that the resistance of adult plants and the resistance of seed coats to mottling are regulated by distinct genes. The identified QTLs on chromosome 17 supplied a novel locus of soybean resistance to seed coat mottling. Using the genetic population resulting from the cross between cultivar soybean and the wild soybean, numerous novel resistance genes have been identified, including the allele that increases alkaline salt tolerance in soybean [29,30]. According to Guo et al., *GmSALT18* and *GmSALT18* conferred salt-tolerant in wild soybean germplasm [31]. In this study, the CSSL population was generated by multiple recurrent crosses [17]. The substituted wild soybean genomic segments could cover about 95% of the genomic information of wild soybean ZYD00006. However, there is still large fragment distribution of the wild soybean genome, and relatively more minute fragments are needed to precisely map the novel candidate genes.

*Glyma.17g238900*, the candidate genes associated with resistance to seed coat mottling determined in this work, encodes a protein homologous to RSS3 of rice. It has been proven that RSS3 of rice functions as a nuclear factor [32]. It lacks the DNA-binding domain but contains the N-terminal domain of basic helix-loop-helix (bHLH) transcription factors. It has been shown that rice RSS3 forms a ternary complex with the class-C bHLH transcription factor bHLH094 and the JASMONATE ZIM-domain (JAZ) protein JAZ9, which represses the expression of jasmonate-induced genes and regulates root cell elongation under saline conditions [18]. Jasmonic acid (JA) and its derivatives, such as methyl-JA and JA-Ile, are crucial phytohormones in plant immune defense against insect, nematode, and microbial pathogens, including bacteria, fungi, and viruses [33,34,35]. It has been found that the JA signaling pathway regulates host resistance to numerous plant viruses, including rice black-streaked dwarf virus (RBSDV), rice stripe virus (RSV), tomato yellow leaf curl virus (TYLCV), rice ragged stunt virus (RRSV), and beet curly top virus (BCTV) [36]. Moreover, the JA signaling pathway displays synergetic or antagonistic crosstalk with other phytohormone-mediated defense pathways, such as salicylic acid (SA) and brassinosteroid (BR) pathways [37,38]. In addition, Yang et al. found that CP of RSV-induced JA signaling has synergetic crosstalk with the RNA silencing pathway through regulation of ARGONAUTE 18 (AGO18) expression, enhancing antiviral defenses of rice [39]. In this work, RSS3L encoded by *Glyma.17g238900* shared a consistent structure with rice RSS3, and transient expression of *GsRSS3L* from wild soybean ZYD00006 inhibited multiplication of SMV in *N. benthamiana*, while *GmRSS3L* from soybean SN14 showed no effect on SMV multiplication. It is possible that RSS3L also regulates the JA signaling pathway and further functions in antiviral defense. However, the extensive mechanisms of antiviral defense mediated by GsRSS3L remain unknown.

Soybean genetic transformation is a valuable tool for the functional study of genes. However, the genetic transformation of soybean remains limited because the common methods employed, such as cotyledonary node-*Agrobacterium*-mediated and somatic embryo-particle-bombardment-mediated transformation, are time consuming and inefficient [40,41]. *N. benthamiana* is the most-applied model plant in plant virology due to its susceptibility to a large number of plant viruses and its great applicability to transient production of proteins via agroinfiltration [42]. Therefore, *N. benthamiana* combined with an *Agrobacterium*-mediated infectious cDNA clone of plant virus has been widely used in studies of plant–virus interactions [42,43,44]. However, the host range of SMV is narrow and most of the SMV strains only systematically infect leguminous plants [45]. According to the reported study by Gao et al., SC7 (a novel SMV strain) could systematically infect *N. benthamiana* from 21 SMV strains in China (SC1-SC21) and construct its infectious cDNA clone [41]. Based on the work of Gao et al., through Agrobacterium-mediated transient co-expression of candidate genes and an infectious clone of SMV SC7 in *N. benthamiana*, *Glyma.17g238900* was identified to be able to inhibit the multiplication of SMV from five candidate genes, indicating that this system can be further used to rapidly screen soybean resistance genes for SMV.

## 4. Materials and Methods

### 4.1. Genetic Materials

In this study, a wild soybean CSSL population consisting of 194 lines and covering 82.6% of the wild soybean genome was used [17]. This population was developed using SN14 as the recurrent parent and ZYD00006 as the donor parent, three back-crosses of selected F1 lines to SN14, and multiple rounds of selfing [17]. A fine genetic bin map composed of 3196 bin markers was constructed for this population (unpublished). The CSSLs and two parental lines were grown in plastic pots with a diameter of 16 cm and a height of 18 cm in a pest-free greenhouse at a temperature of 23 ± 3 °C and a photoperiod of 14/10 h (day/night). Five seeds were planted in each pot and three pots were used for each line.

### 4.2. Virus Culture, Inoculation, and RT-PCR Detection

Prior to the inoculation tests, the N3 strain of SMV, which was used in the current study, was activated and propagated in SN14 after being stored at −80 °C. The inoculum was prepared by grinding symptomatic leaves of SN14 in 0.01 M PBS buffer (1:10 *w*/*v*) and mixed with a small amount of 600-mesh carborundum. By using a paintbrush to gently rub the inoculum on leaves and rinsing with tap water after five minutes, two-week-old soybean plants were inoculated. This inoculation process was repeated once after one week. Then the symptoms on inoculated plants were monitored, followed by calculating the incidence rate of seed coat mottling for each line.

The non-inoculated systemic leaves were collected at 21 dpi and tested by RT-PCR with SMV-specific primers (Appendix A) to confirm the systemic infection of the virus. Total RNA was extracted using TRIzol reagent (Invitrogen, Waltham, MA, USA), while the first-strand cDNA was synthesized using the HiScript II First-strand cDNA synthesis kit (Vazyme Biotech, Nanjing, China). Next, a 20 μL mixture containing 1 μL of cDNA, 10 μL of Premix LA Taq DNA polymerase (TaKaRa, Beijing, China), and 0.8 μL of 10 μM forward and reverse primers was used for the PCR, which was performed in 30 cycles of 98 °C/10 s, 55 °C/30 s, and 72 °C/45 s.

### 4.3. QTL Detection

Based on the genetic bin map of the CSSL population, the incidence rate of seed coat mottling was used for QTL analysis using IciMapping software [46]. An LOD of 2.5 and *p*-value of 0.01 were set as testing criteria. The candidate genes were obtained according to the physical positions of obtained blocks on the Williams 82 soybean reference genome, i.e., GlymaWm82.a2.v1 (https://phytozome-next.jgi.doe.gov/info/Gmax_Wm82_a2_v1, accessed on 3 May 2022).

### 4.4. Transcriptome Analysis

SMV N3 and PBS buffer were used as control treatments while inoculating the SN14 plants. At 3, 6, 12, 24, and 72 hpi, leaf samples from SMV-inoculated and control-inoculated plants were taken. Three biological replicates were collected for each group. Total RNAs were extracted and used for cDNA library construction by the Illumina TruSeq Kit (Illumina Inc., San Diego, CA, USA). The cDNA library sequencing was performed on an Illumina HiSeq 4000 PE 150 platform in Biomarker Technologies (Beijing, China). The raw RNA-Seq reads were mapped to the GlymaWm82.a2.v1 genome using Hisat2 [47]. Fragments Per Kilobase of Transcript Sequence per Million Base Pairs Sequenced (FPKM) values were calculated and standardized using DESeq2 [48]. Furthermore, the read counts for each gene were obtained using HTseq-count [49]. The DEGs compared with the corresponding control treatment at each time point were determined with a false discovery rate (FDR) ≤ 0.01 and the absolute value of log2(Fold-change) ≥ 2 as the threshold.

### 4.5. Prediction and qRT-PCR Validation of Candidate Genes

Based on both the results of QTL detection and RNA-seq, candidate genes related to seed coat mottling resistance were selected from the DEGs localized in regions identified by QTL mapping and employed in subsequent analyses. The qRT-PCR analyses were conducted to validate the expression profiling of candidate genes. From total RNA, first-strand cDNA was generated. qPCR was performed in a reaction mixture of 20 μL comprising 10.0 μL 2 × ChamQ Universal SYBR qPCR Master Mix (Vazyme Biotech, Nanjing, China), 0.4 μL of 10 μM respective upstream and downstream primers, and 1.0 μL cDNA, using the subsequent protocol: initial denaturation at 95 °C for 30 s and 40 cycles of denaturation at 95 °C for 5 s, annealing, and extension at 60 °C for 20 s. Three independent replicates were performed for each sample. *GmUKN1* was selected as a reference gene. The expression level was quantified using the 2^−∆∆Ct^ method. Moreover, statistical analysis was performed by Student’s two-tailed *t*-test using SPSS 12.0.

### 4.6. Agrobacterium-Mediated Transient Expression of Candidate Genes

Using the pEASY^®^-Basic Seamless Cloning and Assembly Kit (TransGen Biotech, Beijing, China), the full-length coding sequence of candidate genes was amplified from SN14 and ZYD00006 by RT-PCR and cloned into the Nco I site of the plant expression vector pCambia3301 under the control of the CaMV 35S promoter. Prof. Zhi Hai-Jian of Nanjing Agricultural University generously provided the full-length cDNA infectious clone of the SMV SC7 strain with infectivity in *N. benthamiana*, which was employed as the virus source [41]. Each plasmid was transformed into *Agrobacterium tumefaciens* EHA105 via electrotransformation. The candidate gene-containing EHA105 isolate and the SMV SC7 infectious clone vector were both resuspended in infiltration buffer (10 mM MgCl_2_, 10 mM MES, and 100 μM acetosyringone, pH 5.6) at an OD600 of 0.5, mixed equally, and kept at room temperature for 3 h before being infiltrated into the leaves of 3-week-old *N. benthamiana* plants. *A. tumefaciens* EHA105 isolate containing an empty vector of pCambia3301 was used as control. Leaf samples were collected at 3, 5, and 7 dpi. The accumulation of viral RNAs was analyzed by qRT-PCR with *NbEF1α* as the reference gene.

## 5. Conclusions

In this study, we identified the chromosome region underlying soybean resistance to seed coat mottling using a CSSL population with wild soybean genomic background. All 18 genes in the mapped chromosome region contained, in total, five DEGs that were induced by SMV, which were identified as potential genes for seed coat mottling resistance. *Agrobacterium*-mediated transient co-expression of candidate genes and full-length cDNA infectious clone of SMV showed that *Glyma.17g238900* from wild soybean ZYD00006 (*GsRSS3L*) could inhibit SMV multiplication in *N. benthamiana*. Hence, it has been suggested that *GsRSS3L* confers seed coat mottling resistance, shedding light on the mechanism of soybean resistance to SMV.

## Figures and Tables

**Figure 1 ijms-23-07577-f001:**
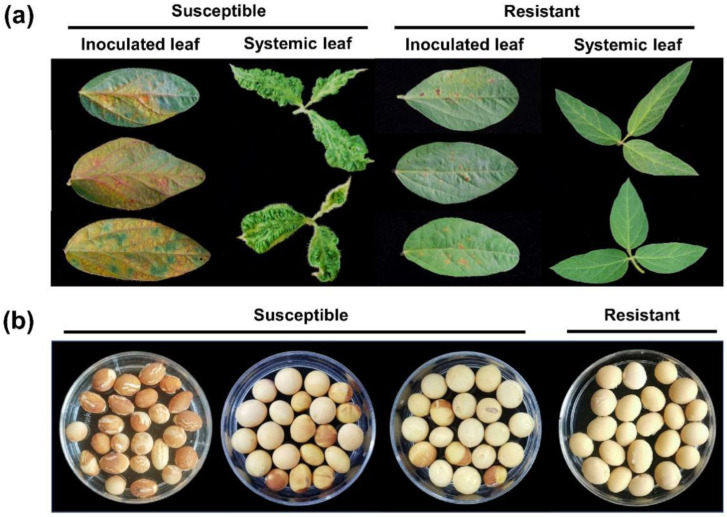
Different disease reactions of chromosome segment substitution lines (CSSLs) after inoculation of SMV N3 strain. (**a**) Leaf symptoms of CSSLs with susceptibility or resistance against SMV N3 infection; (**b**) seed symptoms of CSSLs with susceptibility or resistance against seed coat mottling.

**Figure 2 ijms-23-07577-f002:**
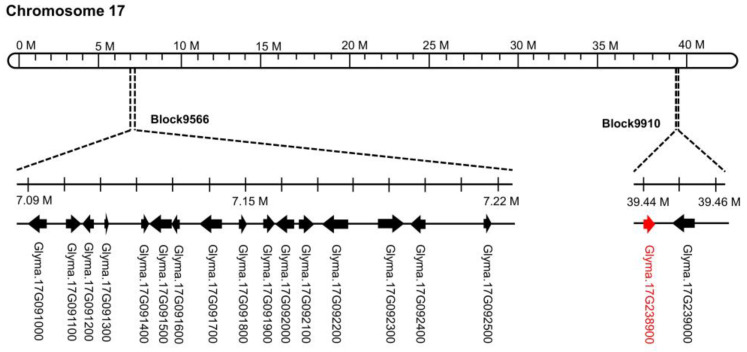
Physical map of the genomic regions associated with resistance to seed coat mottling determined by QTL. The predicted genes in the two blocks are based on the reference genome GlymaWm82.a2.v1 (https://jgi.doe.gov/, accessed on 3 May 2022).

**Figure 3 ijms-23-07577-f003:**
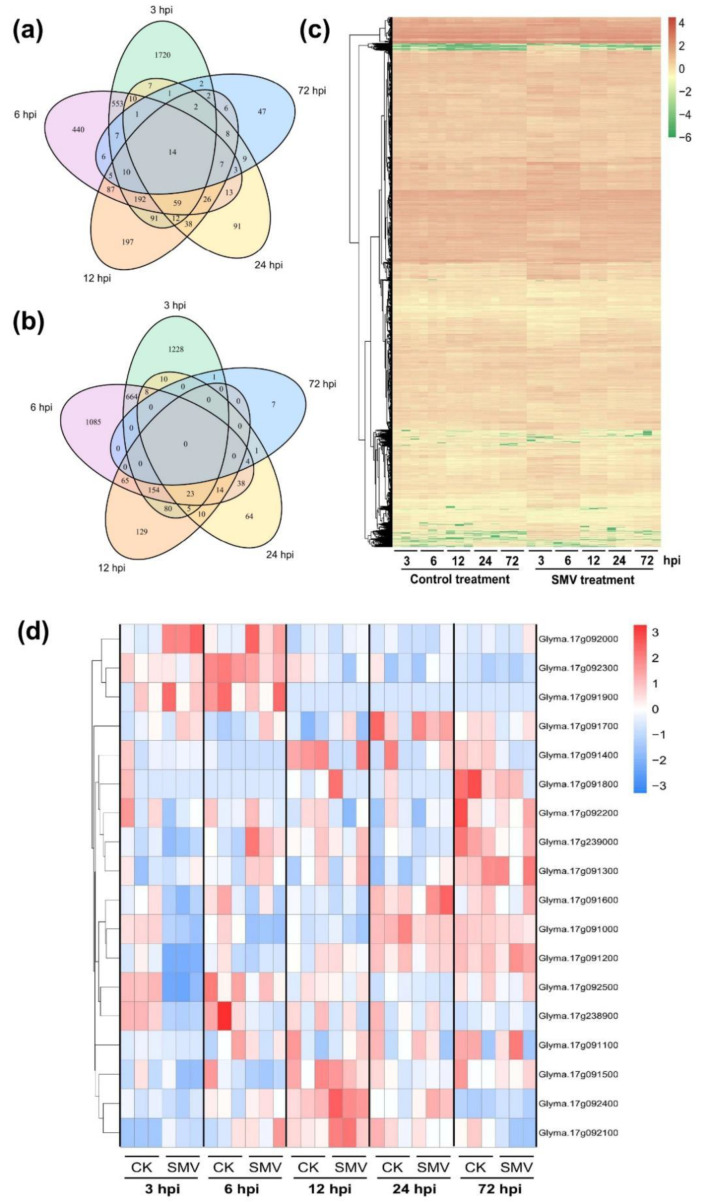
Transcriptome analysis of soybean cultivar SN14 in response to SMV infection. Venn diagram of upregulated (**a**) and downregulated (**b**) differentially expressed genes (DEGs) at 3, 6, 12, 24, and 72 h post-inoculation (hpi); heatmaps of all DEGs (**c**) and candidate genes (**d**) in the regions identified by QTL.

**Figure 4 ijms-23-07577-f004:**
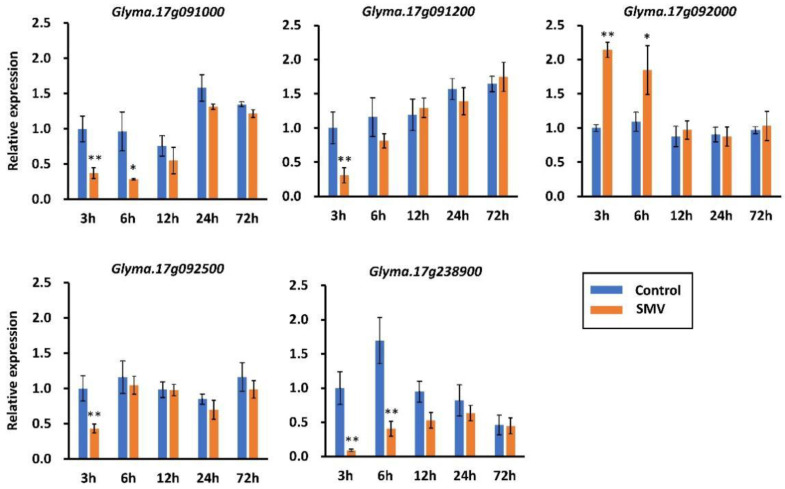
Expression profiling of candidate genes for resistance to seed coat mottling in soybean cultivar SN14 in response to SMV N3 strain. The *y*-axes indicate the relative expression levels between samples infected with SMV and control samples inoculated with PBS buffer; the *x*-axes indicate the time points post-inoculation; the * and ** indicate significant differences at *p* < 0.05 and *p* < 0.01, respectively, based on Student’s two-tailed *t*-test using SPSS 12.0.

**Figure 5 ijms-23-07577-f005:**
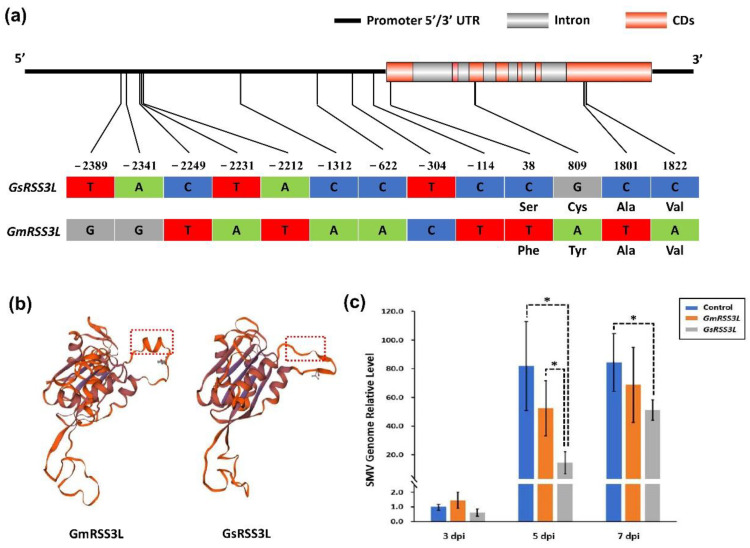
(**a**) Gene structures and single-nucleotide polymorphisms (SNPs) of *GmRSS3L* and *GsRSS3L*; (**b**) predicted three-dimensional structures of protein *GmRSS3L* and *GsRSS3L*, as well as the α-helix region that may be affected by the two nonsynonymous mutants, as indicated by the red box; (**c**) qRT-PCR quantification of SMV genomic RNA in *Nicotiana benthamiana* leaves with transient expression of SMV infectious clone with *GmRSS3L*, *GsRSS3L* and empty vector of pCambia3301 as control, at 3, 5, 7 days post-inoculation (dpi). The * indicates a significant difference at *p* < 0.05.

**Table 1 ijms-23-07577-t001:** The substituted regions associated with resistance to seed coat mottling identified by QTL analysis.

Block	Chromosome	Physical location ^1^		LOD ^2^	PVE (%) ^3^	Add ^4^
		Start	End			
block9566	Chr.17	7,086,191	7,227,595	2.50	4.57	−0.02
block9910	Chr.17	39,435,988	39,459,788	7.90	14.65	−0.05

^1^ Physical locations in reference genome GlymaWm82.a2.v1. ^2^ Logarithm of odds ratio. ^3^ Phenotypic variations explained. ^4^ Additive effects value.

## Data Availability

Data are contained within the article and Appendix A.

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
