# Peer review of "GsRSS3L, a Candidate Gene Underlying Soybean Resistance to Seedcoat Mottling Derived from Wild Soybean (Glycine soja Sieb. and Zucc)"

_ijms, 2022, doi:10.3390/ijms23147577_

Round 1
Reviewer 1 Report
Dear auhtors:
The research is well conducted and the results are sound, therefore I would recommend publication in IJMS. But some parts of the paper must be improved prior to acceptance.
Use of english:
please rephrase and correct grammar in sentences found in:
lines 17-18; lines 204-205 and line 239.
Figure 4: which is the statistical analysis? Students' test? Please mention in the figure legend and include in the materials and methods section a description of how the statistic has been performed and the software used or the calculations performed.
Author Response
Reviewer1
The research is well conducted and the results are sound, therefore I would recommend publication in IJMS. But some parts of the paper must be improved prior to acceptance.
Use of english:
please rephrase and correct grammar in sentences found in:
lines 17-18; lines 204-205 and line 239.
Answer:This is a good comment. We modified the sentences of lines 17-18 and 204-205 and line 209. Thank you very much.
Figure 4: which is the statistical analysis? Students' test? Please mention in the figure legend and include in the materials and methods section a description of how the statistic has been performed and the software used or the calculations performed.
Answer: This is a good comment. The statistical analysis is student’ T-test. We added the related description in the legend and the materials and method section.
Reviewer 2 Report
Comments:
The paper ”GsRSS3L, a candidate gene underlying soybean resistance to seedcoat mottling derived from wild soybean (Glycine soja Sieb. & Zucc)”, by Song et al., is interesting and focuses on a mechanism of soybean resistance to soybean mosaic virus (SMV). The authors followed a strategy using QTL’s which I find adequate.
The paper is clear, well-constructed and the study was well designed and organised. The article shows a lot of good work in my opinion, and it fits well on the special issue of IJMS Special Issue: “Genetics and Novel Techniques for Soybean Yield Enhancement”.
Nevertheless, the text needs and deserves a proper English revision.
The abstract is clear, pointing out the main results. Nevertheless, I advise to rephrase the period “In this study, a chromosome segments substituted lines with wild soybean genomic background were used to identify the chromosome region and mining the candidate genes underlying soybean resistance to seedcoat mottling.” in order to get it clearer (at least, erase the “a” from the beginning of the period, after “In this study,…”).
The Introduction is well organised, and the objectives are clearly stated.
Please correct “To data…” to “To date,…”, on line 54.
Line 57: please rephrase the sentence. Replace “named” by “is named”.
Line 62, change “is” by “are”.
Line 65: replace “have” by “has”.
Line 66:
Results are well and extensively described with the appropriate number of tables and figures and supplementary material, clearly presented and are important to the state of the art in this topic of genetic resistance to seedcoat mottling. The figures/images are of good quality and elucidative.
Please pay attention to this: it does not make sense the separation between the two periods/sentences…(lines 103-105).
“As the substituted chromosome segments of wild soybean can change the soybean resistance to adult-plant resistance and seedcoat mottling. We wish to identify the chromosome region underlying the phenotype difference.”
Please rewrite the two sentences above.
Discussion of the results is also complete and addresses the most important questions.
Nevertheless, the first sentence for me is somewhat awkward:
“In this study, the identified QTLs underlying resistance to seedcoat mottling on the chromosome 17 is a novel resistant locus in soybean.”
At least I would change to “In this study, the identified QTLs underlying resistance to seedcoat mottling on the chromosome 17 permitted to identify a novel candidate resistant locus in soybean.”
In line 205, I would replace the word “underlying” by “determined”.
At the end of Discussion, in order to avoid some misperception, replace in line 251 “In this work” by “In that work” or by “In this work of Gao et al. (2015) …”
I find that the main conclusions at the end of Discussion are missing and for me, it would highlight the quality of the extensive research the authors conducted in this study.
Materials and Methods are appropriate and are well explained and described. The used techniques and strategies are adequate to fully respond to the aim of the study.
Please correct in line 274 “was” to “were”.
The list of references is not extensive, but I think it is satisfactory. The references are well formatted and many are recent!
I congratulate the authors for the good work they present here. All the article needs is a deep english revision.
Author Response
The paper ”GsRSS3L, a candidate gene underlying soybean resistance to seedcoat mottling derived from wild soybean (Glycine soja Sieb. & Zucc)”, by Song et al., is interesting and focuses on a mechanism of soybean resistance to soybean mosaic virus (SMV). The authors followed a strategy using QTL’s which I find adequate.
The paper is clear, well-constructed and the study was well designed and organised. The article shows a lot of good work in my opinion, and it fits well on the special issue of IJMS Special Issue: “Genetics and Novel Techniques for Soybean Yield Enhancement”.
Nevertheless, the text needs and deserves a proper English revision.
The abstract is clear, pointing out the main results. Nevertheless, I advise to rephrase the period “In this study, a chromosome segments substituted lines with wild soybean genomic background were used to identify the chromosome region and mining the candidate genes underlying soybean resistance to seedcoat mottling.” in order to get it clearer (at least, erase the “a” from the beginning of the period, after “In this study,…”).
Answer: This is a valuable comment. We modified the sentences as your suggestion. Thank you very much. It is helpful for our further manuscript preparation.
The Introduction is well organised, and the objectives are clearly stated.
Answer: Thank you very much!
Please correct “To data…” to “To date,…”, on line 54.
Answer: This is a good suggestion, we modified the ‘To date…’ to ‘To date, ….’
Line 57: please rephrase the sentence. Replace “named” by “is named”.
Answer: This is a good suggestion, we modified the ‘named’ to ‘is named’
Line 62, change “is” by “are”.
Answer: This is a good suggestion, we modified the ‘is’ to ‘by’.
Line 65: replace “have” by “has”.
Answer: This is a good suggestion, we modified the ‘have’ to ‘has’.
Line 66:
Results are well and extensively described with the appropriate number of tables and figures and supplementary material, clearly presented and are important to the state of the art in this topic of genetic resistance to seedcoat mottling. The figures/images are of good quality and elucidative.
Please pay attention to this: it does not make sense the separation between the two periods/sentences…(lines 103-105).
“As the substituted chromosome segments of wild soybean can change the soybean resistance to adult-plant resistance and seedcoat mottling. We wish to identify the chromosome region underlying the phenotype difference.”
Please rewrite the two sentences above.
Answer: This is a good comment. We modified the sentences as ‘In the CSSLs, resistance to seedcoat mottling is different, which due to the difference of chromosomal substituted segments with wild soybean genomic information. We wish to identify the chromosome region underlying the phenotype difference.’ We wish the meaning is more clearly.
Discussion of the results is also complete and addresses the most important questions.
Nevertheless, the first sentence for me is somewhat awkward:
“In this study, the identified QTLs underlying resistance to seedcoat mottling on the chromosome 17 is a novel resistant locus in soybean.”
At least I would change to “In this study, the identified QTLs underlying resistance to seedcoat mottling on the chromosome 17 permitted to identify a novel candidate resistant locus in soybean.”
Answer: This is a valuable suggestion. Thank you very much! We modified the sentence as your suggestion.
In line 205, I would replace the word “underlying” by “determined”.
Answer: This is a valuable suggestion. We replaced the word “underlying” by “determined”.
At the end of Discussion, in order to avoid some misperception, replace in line 251 “In this work” by “In that work” or by “In this work of Gao et al. (2015) …”
Answer: This is a valuable comment. Here, we descripted sentence might make misunderstanding. As the Work of Gao et al. (2015) identified the SC7 can infect soybean. So based his work we construct a SC7-GFP clone. We wish keep our original sentence ‘In this work’.
I find that the main conclusions at the end of Discussion are missing and for me, it would highlight the quality of the extensive research the authors conducted in this study.
Answer: this is a good comment. We added the conclusion section as ‘In this study, we identified the chromosome region underlying soybean resistance to seedcoat mottling using a CSSLs population with wild soybean genomic background. A total of five DEGs induced by SMV were found from all of the 18 genes located in the mapped chromosome region, and were determined as candidate genes for resistance to seedcoat mottling. Agrobacterium-mediated transient co-expression of candidate genes and full-length cDNA infectious clone of SMV showed that Glyma.17g238900 from wild soybean ZYD00006 (GsRSS3L) could inhibit SMV multiplication in N. benthamiana. We proposed that GsRSS3L may confer the resistance to seedcoat mottling.
’
Materials and Methods are appropriate and are well explained and described. The used techniques and strategies are adequate to fully respond to the aim of the study.
Please correct in line 274 “was” to “were”.
Answer: This is a good comment we modified the “was” to “were”.
The list of references is not extensive, but I think it is satisfactory. The references are well formatted and many are recent!
Answer: Thank you very much!